# Innovative Firmware Update Method to Microcontrollers during Runtime

Bernardino Pinto Neves [1], Victor D. N. Santos [2,3] and António Valente [1,4,*]

1 Engineering Department, School of Sciences and Technology, University of Trás-os-Montes and Alto Douro (UTAD), Quinta de Prados, 5000-801 Vila Real, Portugal; bernardino.p.neves@gmail.com
2 Polytechnic Institute of Coimbra, Coimbra Institute of Engineering, Rua Pedro Nunes-Quinta da Nora, 3030-199 Coimbra, Portugal; vsantos@isec.pt
3 INESC Coimbra, DEEC, Polo II, 3030-290 Coimbra, Portugal
4 INESC Technology and Science, 4200-465 Porto, Portugal
* Correspondence: avalente@utad.pt

**Abstract:** This article presents a new firmware update paradigm for optimising the procedure in microcontrollers. The aim is to allow updating during program execution, without interruptions or restarts, replacing only specific code segments. The proposed method uses static and absolute addresses to locate and isolate the code segment to be updated. The work focuses on Microchip's `PIC18F27K42` microcontroller and includes an example of updating functionality without affecting ongoing applications. This approach is ideal for band limited channels, reducing the amount of data transmitted during the update process. It also allows incremental changes to the program code, preserving network capacity, and reduces the costs associated with data transfer, especially in firmware update scenarios using cellular networks. This ability to update the normal operation of the device, avoiding service interruption and minimising downtime, is of remarkable value.

**Keywords:** firmware update; partial update; runtime; internet of things; microcontrollers

## 1. Introduction

In electronic devices that incorporate microcontrollers, it is common to implement firmware update mechanisms to correct errors and make new services available after the product has been launched. Firmware updates often involve risks related with downtime, failure of the update itself, and costs associated with communications to support those updates. The article aims to address these limitations by presenting an innovative firmware update method that minimises or eliminates downtime and optimises the data to be updated. Despite the importance of this topic, there is little research into efficient firmware update methods that minimise or eliminate downtime. There are devices for which interruption of operation is critical, for example, the digital control of the power supply of a data centre (or other critical system) in a non-redundant configuration. In this scenario, firmware updates on the power supply unit can lead to temporary service interruptions [1]. Kilpeläinen [2] presents an innovative method for dynamic firmware updates, addressing updates without the need to reboot the device and modify the program code during execution. With regard to the efficient use of the communications channel, the literature refers to methods for optimising the data transmission to be updated. Bogdan [3] focuses on optimising data transmission in firmware update processes, detailing the concept of delta transmission and its combination with data compression. That work is based on the use of opcodes instead of addresses, offering an innovative perspective to efficiently transmit the updates. The system inactivity time present in the aforementioned methods, which assume a reboot after the update, led to the proposal of an innovative firmware update method based on block updates, with the aim of replacing specific code segments the program's memory, which is done during runtime and without the need for a reboot.

The originality of this study lies in the innovative approach of updating firmware by blocks, enabling an efficient and secure implementation while minimising negative impacts on system operations. This new method is expected to significantly reduce downtime and the use of communication channels. A circuit with a `PIC18F27K42` microcontroller [4] was developed to validate the method. The firmware that comprises the applications and the update process was initially uploaded to that circuit using a `RS232` serial channel and a serial terminal. The article is organised in sections. Section 2 describes several similar related studies. Section 3 gives a detailed description of the implemented block oriented firmware update method and the assumptions that allow the method to be successfully replicated. The communications protocol used to perform the update file transfer is also described as well the update process. In Section 4, the results obtained are described. Section 5 presents the main conclusions derived from the findings of this study.

## 2. Related Work

Several notable studies were analysed related to the firmware updates management, optimisation of the update files transmission, and improving the process of writing to the microcontroller's program memory. In the field of firmware update management, Mahfoudhi [5] describes an over-the-air firmware update management model for `NB-IoT` networks as the number of end devices increases significantly, seeking improvements in flexibility, installation time, efficiency, and cost reduction. In a similar context, Frisch [6] proposes a set of models and rules for the firmware update process based on secure distribution and automatic installation mechanisms. Kachman [7] addresses energy efficiency and its impact on firmware update processes as well as explores the evolution of this method based on delta transmission. In the area of optimising the transmission of update files, several significant studies stand out. Wee [8] presents a methodology for transmitting update files that is based on the differences between the new and old firmware, with the aim of optimising the firmware update process. Moreover, a high speed compression and decompression algorithm to significantly speed up the update time is described. Ji [9] refers to a study that focuses on the incremental firmware update method by modules. This method is based on assigning memory zones to each module and introducing the concept of static allocation of functions and relevant security considerations. This innovative approach improves the efficiency and security of firmware updates. Regarding the optimising of the writing process of to the microcontroller's program memory, several studies have made significant contributions. Jisu Kwon [10] presents a method of updating the microcontroller's program memory based on updating by functional blocks. This makes possible a partial update of the program memory instead of completely rewriting it, avoiding downtime during the update process. Xia [11] presents the concept of function addressing by means of a module orientated programming model. In this model, the code is organised around modes and modules for a generic dispatching procedure. Xia also introduces the concept of multimode application management, grouping together applications with similar behaviour and analysing performance evaluation techniques and metrics. Dhakal [12] presents an architecture based on delta updates and incremental mode for large scale `IoT` systems and refers to the ability to verify firmware integrity, highlighting the advantages of delta updates and identifying scenarios in which this method may not be efficient. Sun [13] reveals the limits of conventional firmware update methods and proposes a method that uses partial updates, optimising the lifetime of program memory. This method is based on partitioning the program memory into several sections, updating only the relevant section, and classifying each partition as a component. The study addresses security mechanisms, such as encryption, signing, and validation before and after the update, as well as solutions for the static allocation of functions in scenarios where the function addresses are different between the two firmware versions; in addition, the update method is based on packets that include the functions or modules to be updated, and the study presents a statistical analysis of update times as a function of the transmission channel. Kwon [14] proposes partitioning the firmware into functional blocks, introducing the concept of a function

map. The method aims to update only the functional blocks with differences, reducing the use of program memory, energy consumption, and update time. This involves sending a functional block, where the updating application checks for differences and updates only what is necessary, then updating the function map to reflect the new state. Baldassari [15] explores delta firmware updates in scenarios with bandwidth constraints by updating only small memory files of the firmware. The study details the delta update process, which requires one application to build the delta file and another to rebuild the new firmware from the received deltas. Although this approach offers the advantage of updating the firmware with small memory files, it also has disadvantages, such as greater complexity compared to traditional methods, a higher probability of failure, and the need to keep a copy of the original version of the firmware in the microcontroller. In addition, it requires substantial resources on the microcontroller side, including memory and processing to handle delta updates and corrections.

## 3. Method Development

The underlying idea of the proposed new method consists of the `Non Volatile Memory (NVM)` controller usage to directly update parts of the existent program code. The `NVM` controller is a hardware resource present in the majority of microcontrollers that is responsible for the management of non-volatile memory—also known as flash memory—the type of memory that retains data even when the microcontroller is turned off. The above mentioned `NVM` controller acts over the available flash memory blocks allowing one to read, write, and erase the existing data in memory. The use of this `NVM` controller allow us to update the existent firmware during runtime in the same way we can read and write `NVM` user data without compromising the operation of the applications. Consequently, an update task application is added that aims to receive the data blocks associated with the code of a particular application and update them in the flash program memory, as illustrated in Figure 1.

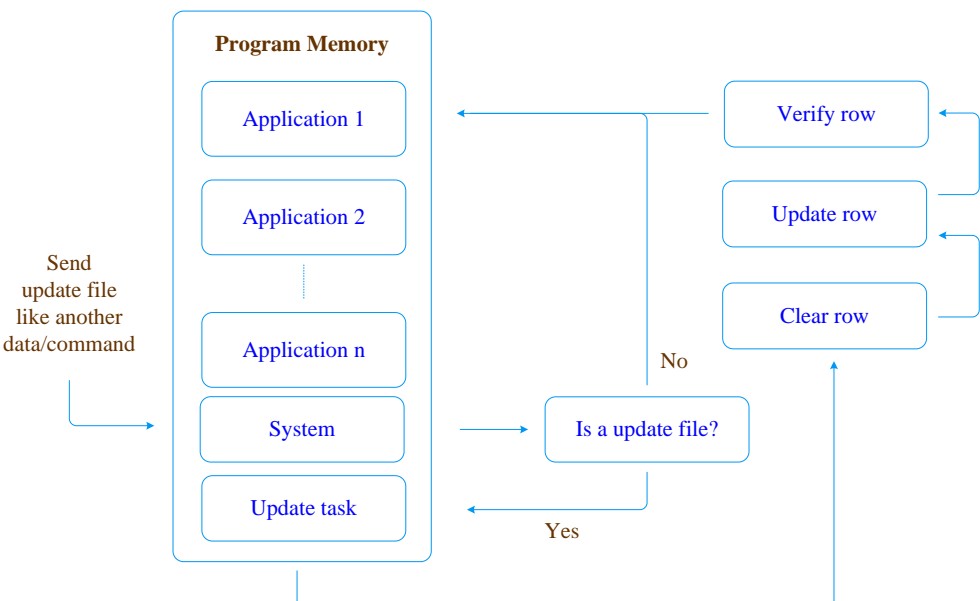

**Figure 1.** Runtime firmware updates method.

The non-volatile memory of a microcontroller is usually segmented or organised into several sectors, most of them devoted to the program memory. The program memory can be configured with different partitions, sizes, and write protection attributes. These partitions can be configured to implement the boot area, the application area, and the user memory data. In this paper, a `PIC18F27K42` microcontroller is used as a testbed platform to validate the proposed techniques. This microcontroller has a non-volatile memory control mechanism that uses an internal timer and voltage generator to perform writing

operations. Reading program memory is executed byte by byte. The writing process is, however, more complex, as it requires the operation to be performed on a row of bytes. The content of this row must be previously erased or available for writing if it is its first use. The writing operation also requires that a write unlock sequence be activated [4]. Writing or erasing program memory will halt the microcontroller central processing unit CPU, making it impossible to execute instructions from the memory row that is being erased, as the microcontroller CPU is blocked until the process is completed [4]. For the above mentioned PIC18F27K42, the measured erasing and writing procedures take 10 ms per row. Table 1 illustrates the size and number of rows [4].

**Table 1.** Size and number of rows, PIC18F27K42.

| Description | Value | Units |
|:---:|:---:|:---:|
| Erase Row Size | 64 | Word |
| Length Row | 128 | Byte |
| User Rows | 1024 | Byte |

The program memory read operation does not modify data; therefore, it is very simple to carry out, simply defining the memory area to be accessed. To complete this operation, we need to previously select the program flash memory and set the address to be read using the TBLPTR register, then read the contents of that position. Note that the reading is performed byte to byte, but each program memory position has a size of two bytes; therefore, it is necessary to increment the pointer of the reading table TBLRD for each byte read. The result is in the register TABLAT: the first byte corresponds to the less significant byte and the second to the most significant byte of the specified memory position content [4]. To read the contents of a particular program memory address, the following sequence of operations must be completed, as illustrated in the flowchart of Figure 2.

**Figure 2.** Reading the contents of program memory PIC18F27K42.

The write operation follows the same principle as the read operation, but operates over rows instead of bytes. The write operation is performed on an entire row, but it is implemented byte by byte [4]. As a recommended practice, in a write operation in which only part of the row is changed, it is suggested that the row be read and stored in volatile memory RAM before being erased. The copied row is then updated with the portion of the data that differs from the original version. Finally, the NVM row should be deleted and rewritten with the updated version. For the writing process to be successful, we must first make sure that the row is available for writing; in other words, the row is

formatted. Thereafter, it is necessary to define the NVM area to be used for writing, where through the TBLPTR register we define the address we want to write; as with reading, the writing is also done byte by byte, and, in the writing process, the least significant byte is copied to the register TABLAT followed by the increment of the writing table TBLWR. That process is repeated for the most significant byte. After copying the row, the next step involves activating the NVMCON1bits.WREN write permission bit as well as selecting the NVMCON1bits.FREE write bit command, followed by sending the write unblock sequence to the NVM. The actual write is initiated by activating the NVMCON1bits.WR bit [4]; see the flowchart in Figure 3.

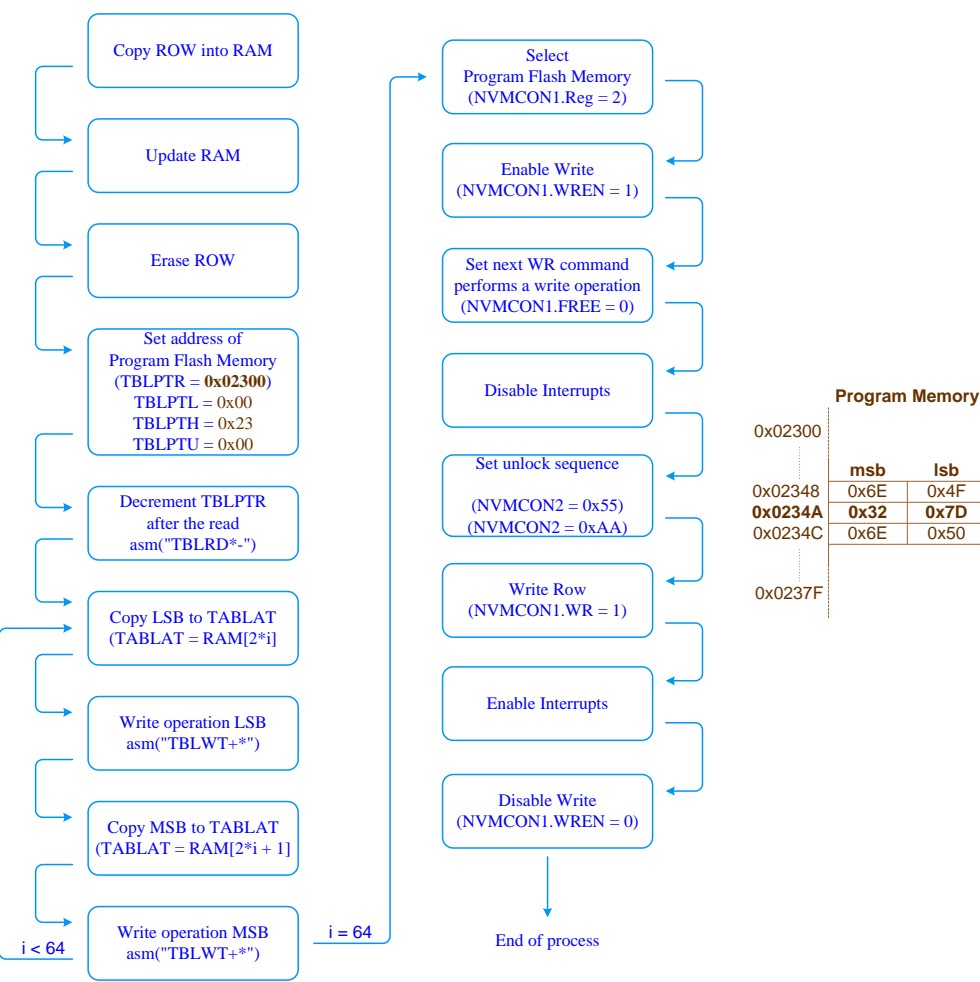

**Figure 3.** Writing process to program memory PIC18F27K42.

To erase a row of non-volatile memory, a specific NVM controller command is used devoted for that purpose. The FREE bit of the NVMCON1 register, if enabled, indicates that on the next enable the WR bit of the same register will erase the row specified by the address contained in the TBLPTR register. Moreover, it is necessary to previously unlock a specific range of rows to accommodate the program code and thereafter complete the erase procedure [4], as depicted in the flowchart in Figure 4.

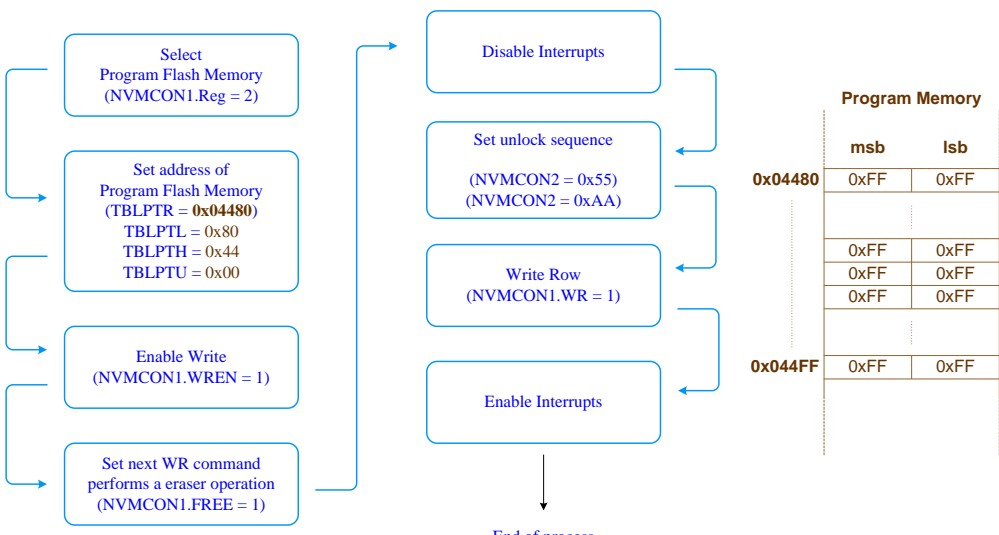

**Figure 4.** Erasing a row of program memory `PIC18F27K42`.

The `NVM` memory locking mechanism prevents unintended self-write programming or erasing. Thus, to promote memory integrity, any write and erase operation performed by the `NVM` controller must be preceded by an unlocking process. This process must be executed sequentially and without interruptions. If the sequence, for some reason, is interrupted, the writing or erasing process is cancelled [4]. To implement this method successfully, two non-mandatory but highly recommended requirements must be met to facilitate its implementation. The first one concerns the static and absolute allocation of the functions. Typically, a compiler, in order to optimise the space of the memory of the program, leans all the code to minimise the used memory space, making it more difficult to identify the location of the block of code that will need to be updated. By allocating the function's code in a static and absolute way, an absolute reference of the location of each function of program is set, facilitating the identification of the code block in an `Intel Hex` file (see Figure 5).

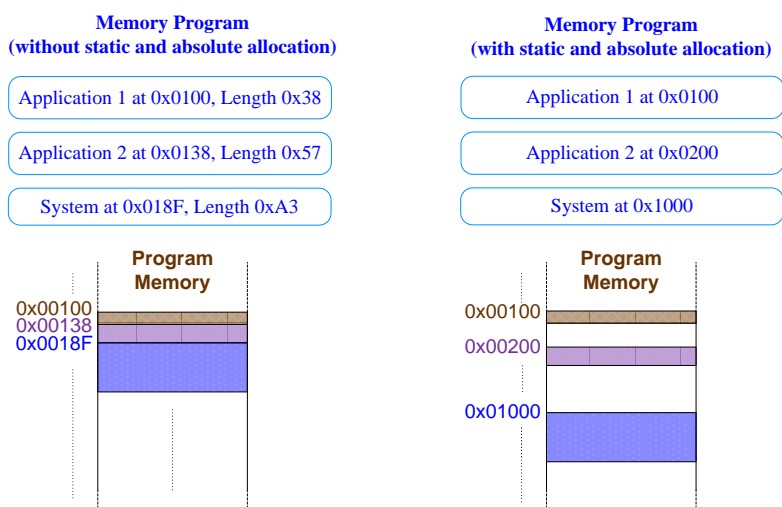

**Figure 5.** Example of memory allocation with and without static and absolute allocation.

The usage of static and absolute function allocation also improves the code organisation. Without static and absolute allocation, even small changes in source code can result in a hex file completely reformulated by the compiler. The usage of static and absolute

allocation avoids major changes. Now, small code changes in specific functions will only affect the associated allocated memory areas, as illustrated in Figure 6.

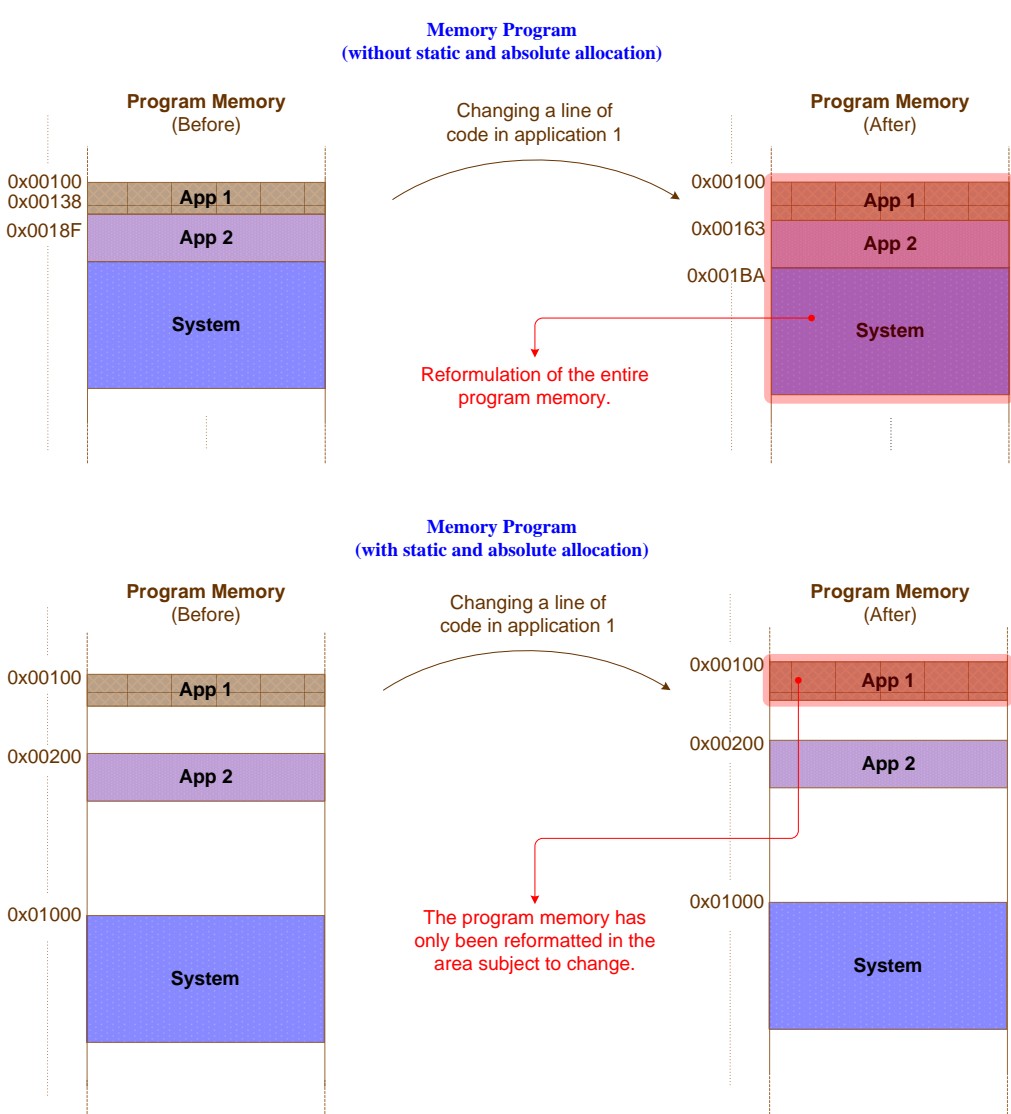

**Figure 6.** Result of the hexadecimal file with the change of only one line of code.

Static and absolute allocation of functions requires well designed system architecture and a complete knowledge of the program's memory map in order to avoid overlaps between the functions or applications code blocks. In order to prevent an accidental overlap of two or more functions, the compiler warns us by displaying a message with the functions that are at stake, promoting the necessary changes in the memory map. The following error message was generated by the compiler under the above mentioned conditions [16].

**error: (596)**
**segment "_Reset_CNT_TMR_text" (19574-195A3)**
**overlaps segment "_TMR0_Interrupt_Handling_text" (194F6-1958F)**

The second requirement concerns the size allocated to each function, which must be an integer multiple of row size; in the considered microcontroller, that size is equal to 128 bytes [4]. An example of a program memory map is depicted in Figure 7.

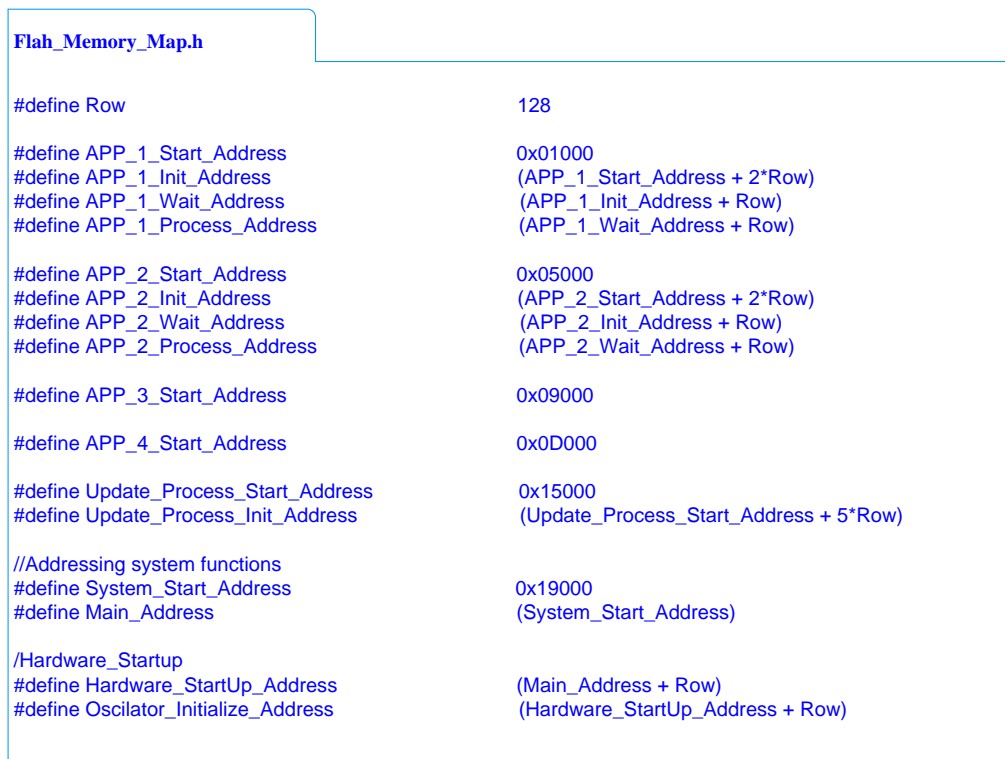

**Figure 7.** Example of program memory map.

The following step, after the program memory map definition, comprises setting the function's indexes addresses in the above mentioned range. To allocate a function in a static and absolute way, one simply needs to add before the function name the method `__at(address);` from here, the compiler will place that function in that specific address, as illustrated in the following function prototype.

**void __at(APP_1_Start_Address) App_1(void)**

To validate this method, a testbed was developed comprising a circuit board with the microcontroller, two push buttons, and an `ICSP` header (depicted in Figure 8).

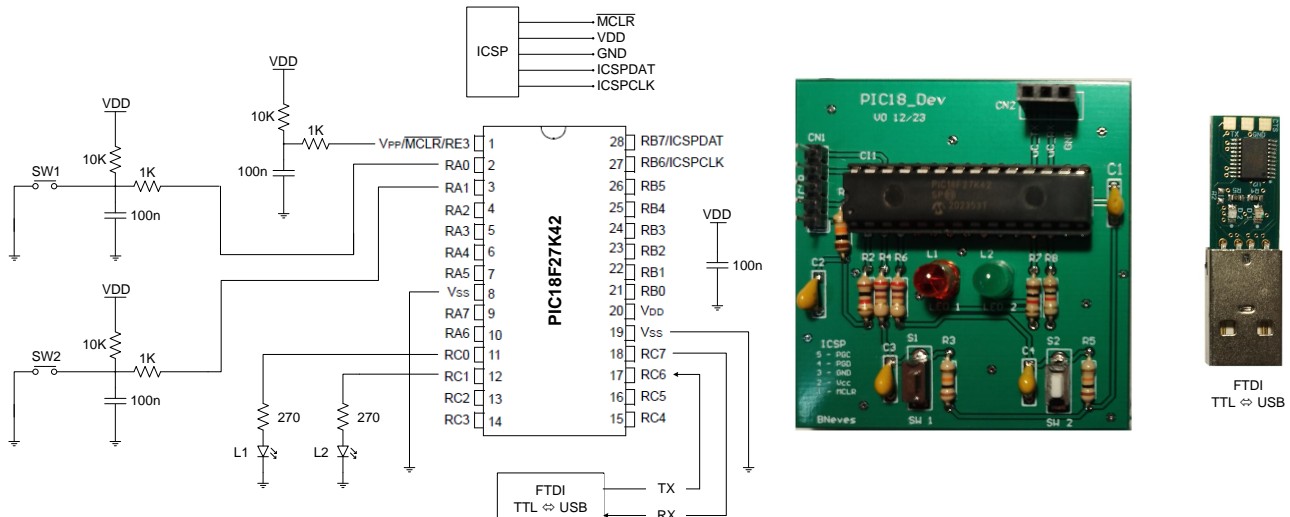

**Figure 8.** Layout of the circuit implemented to validate the method.

The firmware project comprises three applications: two similar applications associated to different hardware resources, in this case push buttons, and In Application Programming

(IAP) that performs an update of the firmware by means of a runtime self programming process. The first application prints in the serial port the message ''`Button 1 has been pressed`'' when button 1 is pressed. Similarly, the second application prints the message ''`Button 2 has been pressed`'' in the serial port when button 2 is pressed. These messages are defined and saved in the microcontroller flash memory. Figure 9 illustrates the flowchart of the implemented program.

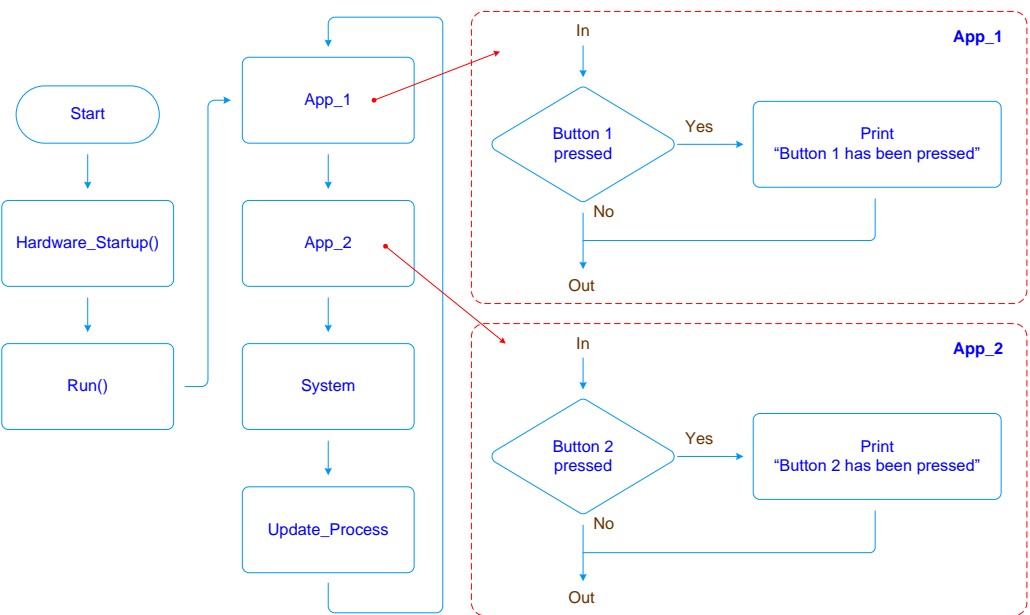

**Figure 9.** Implemented system and applications used to validate the proposed method.

After executing a firmware upgrade operation, it is intended to update the message printed by the first application from ''`Button 1 has been pressed`'' to ''`This string has been changed by update at run time`'', whenever the hardware button 1 is pressed (see Figure 10).

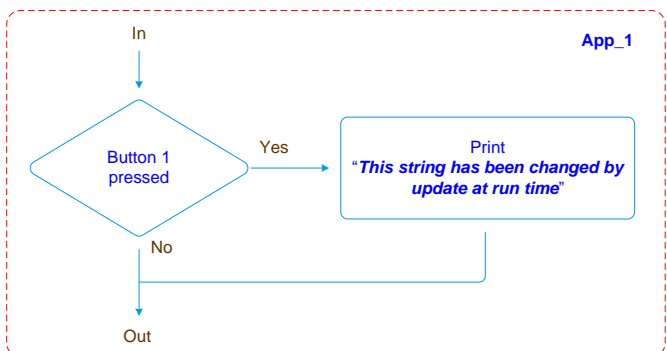

**Figure 10.** Proposed amendment for App_1.

From the analysis of the compiled program code, it can be seen where each function of the first application is allocated in the program memory (see Figure 11 and example of program memory map in Figure 7).

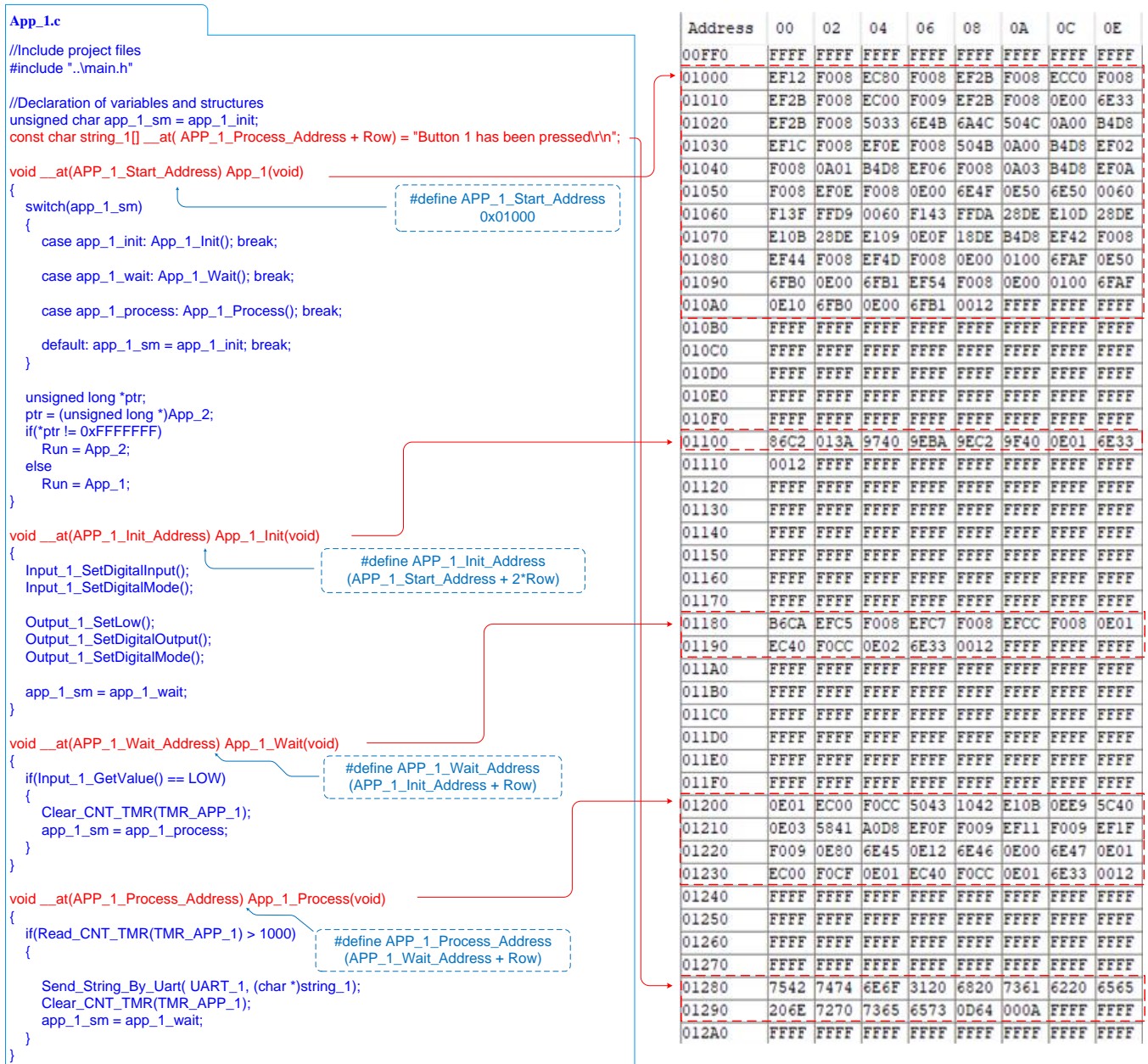

**Figure 11.** Sample code and location of functions in program flash memory.

Additionally, it is also possible to identify and locate application 1 in the hexadecimal file generated from the compiler (see Figure 12).

From the analysis of the modified program hexadecimal file, it can be concluded that only a well defined area of the program memory was changed; all the remaining program memory stays intact. Figure 13 presents the original and upgraded code versions of the aforementioned application 1, demonstrating the code blocks that have been removed on the original version and the ones that have been inserted on the modified one.

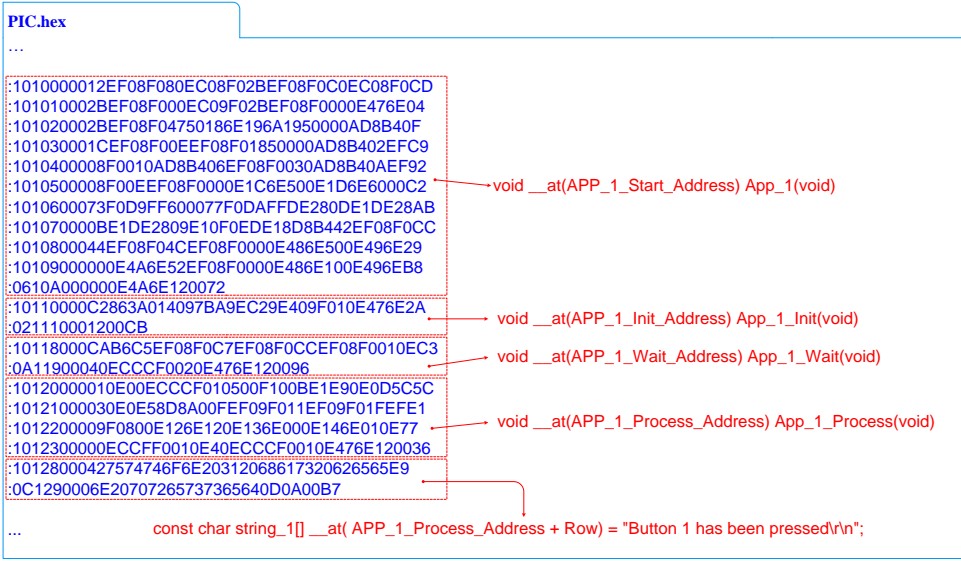

**Figure 12.** Location of functions in program memory in the hexadecimal file.

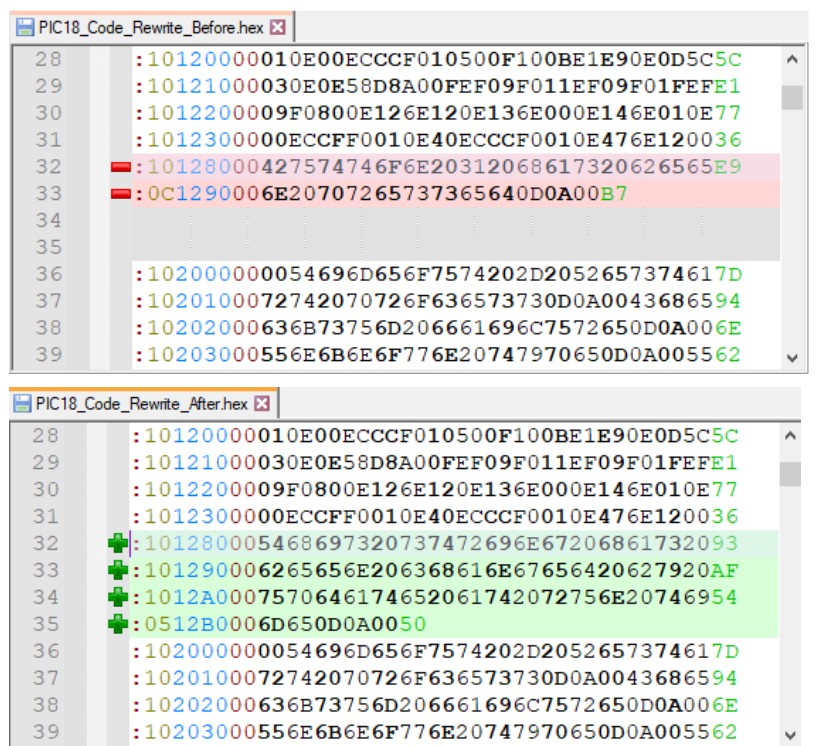

**Figure 13.** Changed program memory area viewed from hexadecimal file.

Using the static and absolute function allocation allows one to control and manipulate the entire program memory, making the updating task easier and keeping the firmware update circumscribed to a well defined block of program memory between addresses 0x00001280 and 0x000012B0. The update process consists of receiving a hexadecimal file in the `Intel Hex File` format [17] over the serial channel. However, as only one block of the program's memory is to be updated and the hexadecimal file is not formatted to send a single block but the entire file, some changes need to be made so that the update process application can interpret the file correctly. Those changes include the addition of a start file, followed by the most significant word of the address and the end of file, as illustrated in Figure 14.

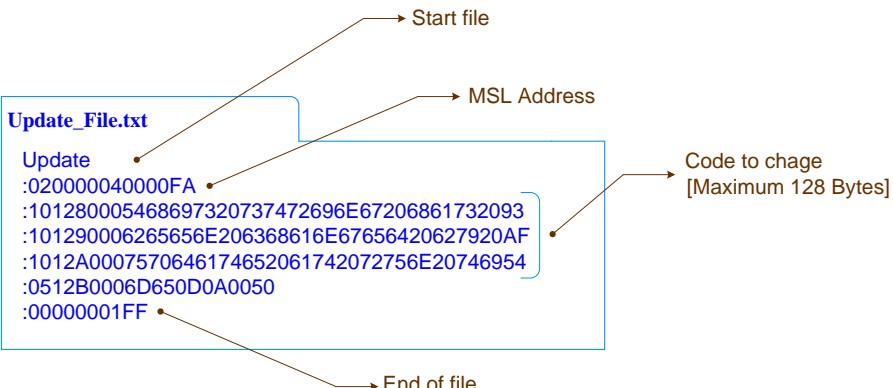

**Figure 14.** Update file in `Intel Hex File` format, adapted to the application.

The `Intel Hex File` format is one of the formats used to update microcontrollers' firmware, but there are also other possible formats, such as the binary `.bin` file. The `Intel Hex File` format is characterized by the lines being in hexadecimal format; all the lines start with the character **':'** followed by the data field length, start address, data type, the associated data (for each specific data type), and, finally, the error control checksum mechanism [17].

Figure 13 depicts the hexadecimal file obtained from the compiled modified code, which is sent to the microcontroller according to the aforementioned Intel Hex File format described in Figure 15 and Table 2. As explained previously, with the inclusion of all the fields, the file sent to the microcontroller is the one presented in Figure 14. The update file results from the extraction of a block of program memory of the hexadecimal file with the updated code; the file is started with a start file named `Update`, followed by the most significant word of the address, the data to update, and, finally, the end of file indicator. The update process application is responsible for the file receiving and processing. The initial state of the update process app waits for the reception of a start file, "Update" string, to proceed to the data acquisition state. In this state, the process waits until it receives a complete record and verifies its integrity using the checksum mechanism. If the line is valid, the process thereafter extracts the address, the type, and the data contained in the line. Depending on the type of data, the process reacts accordingly. For `Extended Linear Address` type, the MSW of the address is defined; if the type is ''`Data_Record`'', it updates the LSW of the address and copies the data to a process buffer; finally, if the type is `End of file`, the process proceeds to the next stage, updating the program memory block. First, it copies the area of the program memory block to be updated to volatile memory `RAM` for final verification purposes of the update integrity; in the next operation, it erases the memory block to be updated, followed by updating with the data received by the update file; finally, a verification is performed between the data in the update file received and the data stored in the updated memory block. The update process application can be seen in the flowchart in Figure 16.

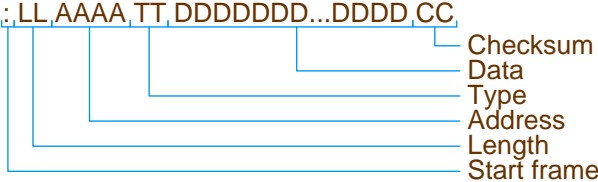

**Figure 15.** `Intel Hex File` record format [17].

**Table 2.** Line fields structure, `Intel Hex File` format [17].

| Field | Designation |
|---|---|
| Start frame | Record start character |
| Length | Two ASCII digits to specify the record data field size |
| Address | Four ASCII digits to define the starting address of this data record. |
| Type | Data type:<br>0—Data record;<br>1—End of file record;<br>2—Extended segment address record;<br>4—Extended linear address record. |
| Data | Data bytes. |
| Checksum | Two ASCII digits representing the checksum calculated as 2s complement of all preceding bytes in data record except the colon. |

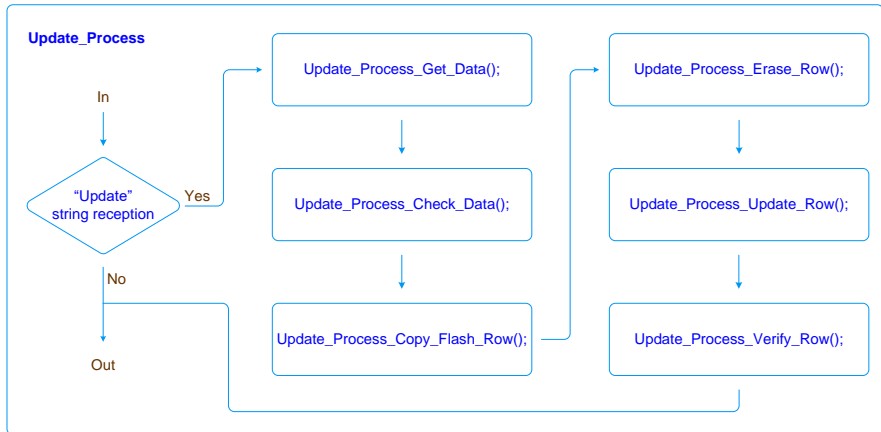

**Figure 16.** Updating process application state diagram.

If the process was completed successfully, it reports '`Update success`'; otherwise, it reports '`Update failure`' through the serial channel. Figure 17 illustrates the update file transfer protocol implemented between the host and the device microcontroller.

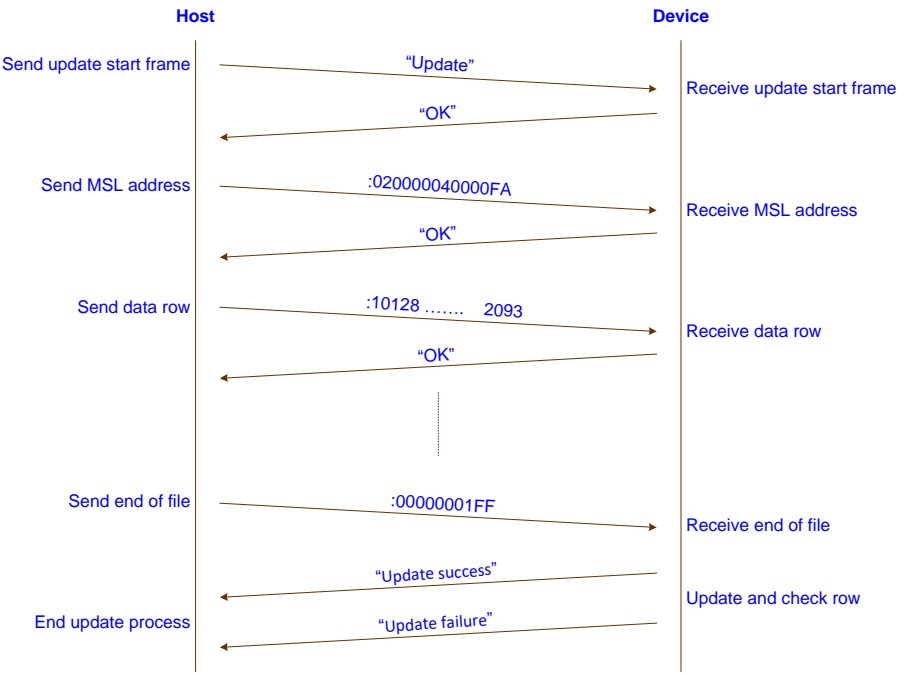

**Figure 17.** Implemented update file transfer protocol.

## 4. Results

This section presents the results obtained by the proposed novel firmware update method during runtime. This method was validated using the previously presented example program and testbed. First, an Intel hex file was sent to the `PIC18F27K42` microcontroller using an RS232 terminal to change program memory, updating the message that occurs when the push button is activated. The update process application receives and verifies the integrity of the Intel hex file, producing the desired modification in one particular block of the flash memory (see Figure 18). The updated applications now operate accordingly with the performed changes. The application assigned to button, 1 instead of printing the message ''`Button 1 has been pressed`'' on the serial port, starts to print a different message: ''`This string has been changed by update at run time`''. It is also possible to verify through the terminal log time that the update completion time took around 63 ms, which is the expected value for an update with a size of 128 bytes. The 63 ms corresponds to about 52 ms spent in the transmission (about 200 bytes at a rate of 38.4 kbps), 10 ms in the block update [4], and about 1 ms in the update process. One of the main contributions of this study is the significant reduction in downtime during the update process as well as the elimination of the need for rebooting the end device after the update. Moreover, this method aims to overcome some limitations associated with the delta firmware update method described in [3,12,15], namely, the requirement to reconstruct the firmware from the deltas, leading to resource savings and process simplification.

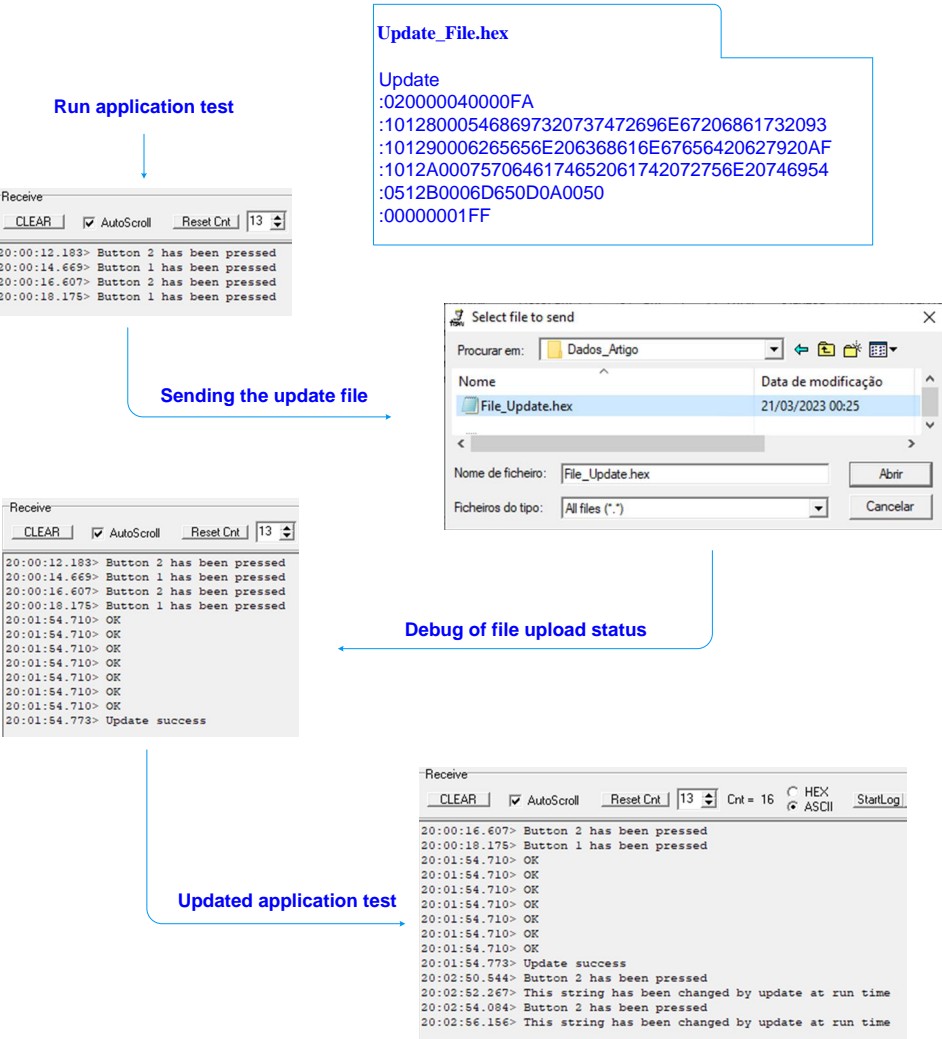

**Figure 18.** Microcontroller update during runtime.

## 5. Conclusions

In this paper, a new firmware update method for microcontrollers is presented, implemented, and validated. This new method differs from existing ones because it allows for updating only specific code lines, blocks, or functions instead of replacing the entire program during runtime. This method is suited to band limited channels that take into account the attained reduction on the amount of data transmitted. The proposed update procedure offers additional advantages, such as a reduced downtime, less than 10 ms, and good recoverability in a failure scenario.

The planned method also presents some limitations; the update process was designed to update only up to eight rows (1024 bytes' maximum), so it is therefore impossible to update the entire program memory at once.

This firmware update method is also incompatible with operating systems and/or intermediate hardware abstraction layers; it requires full control over all functionalities. Moreover, under a power failure event, the success of the update process is not guaranteed. Thus, it is advisable to include a supercapacitor-based backup power circuit to maintain module power and the upgrade process integrity.

This method was successfully and easily replicated on several microcontrollers, such as the MSP430, STM8, STM32, ATtiny, ATmega, SAMD21, and PIC32. This observation emphasises the feasibility and applicability of the method on a broad set of microcontrollers, thus increasing the scope of its potential usefulness. Future advances on the proposed method must consider the inclusion of radio transmission, using LoRaWAN or available cellular networks, to send the update file to remote sensor end-devices. An automated process to manage the partitioning of program memory and assign to each specific function an area of appropriated size based on its likelihood of being updated will also be investigated in the future. In conclusion, this article leaves an open door to a new generation of firmware updates for microcontrollers.

**Author Contributions:** Conceptualization, A.V.; Formal analysis, V.D.N.S.; Investigation, B.P.N.; Methodology, B.P.N., V.D.N.S. and A.V.; Resources, V.D.N.S.; Writing—original draft, B.P.N.; Writing—review and editing, V.D.N.S. and A.V. All authors have read and agreed to the published version of the manuscript.

**Funding:** This research received no external funding.

**Data Availability Statement:** Data are contained within the article.

**Acknowledgments:** This work was financed by National Funds through the Portuguese funding agency, FCT—Fundação para a Ciência e a Tecnologia, within project LA/P/0063/2020. DOI 10.54499/LA/P/0063/2020 | https://doi.org/10.54499/LA/P/0063/2020. This work was supported by the Portuguese Foundation for Science and Technology under the project grant UIDB/00308/2020 with the DOI 10.54499/UIDB/00308/2020 | https://doi.org/10.54499/UIDB/00308/2020.

**Conflicts of Interest:** The authors declare no conflict of interest.

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
