# Peer review of "Innovative Firmware Update Method to Microcontrollers during Runtime"

_electronics, doi:10.3390/electronics13071328_

Round 1

Reviewer 1 Report

Comments and Suggestions for Authors

The article "Innovative Firmware Update Method to Microcontrollers During Runtime" introduces a approach to firmware updates in microcontrollers, promising to change the field by addressing the longstanding issue of downtime and system restarts during updates. However, here are some suggestion for the improvement of the paper:

The Abstract must have the following logic: Purpose; Design /methodology / approach; Findings; Practical implications; Originality/ value;

Its introduction is lacks bibliography. There are only two authors cited! The literature GAP needs to be clarified and reinforced. I recommend reorganization of the introduction, in the following manner:

1-      Make a frame for the reader

2-      Issues of the topic under analysis

3-      Evidence of the GAP of the literature based on the literature,

4-      Purpose of the study

5-      Originality of the study

6-      What are the expected results (to captivate the reader)

7-      The last paragraph, that already exists, should briefly describe what the reader can read in the following sections.

Literature section is correctly written.

In discussing the results, the authors do not confront them with the literature.

Here are some questions that could further expand the relevance of the paper within the discussion section:

1-      How do the authors ensure the integrity of firmware updates during program execution on the microcontroller?

2-      How practical and efficient is this method in real-world scenarios, especially considering its focus on the PIC18F27K42 microcontroller?

3-      How does the method behave in different update scenarios, and in which situations could it be most beneficial?

4-      How does the new method address potential challenges such as security, compatibility, and performance during firmware updates?

5-      Have the authors explored and considered possible drawbacks or weaknesses of the new method, and how do they plan to address them in the future?

6-      How applicable is this method to other microcontrollers or similar systems, besides the Microchip PIC18F27K42 microcontroller?

7-      What are the next steps in researching or developing this method, and how could it be further improved in the future?

And the conclusions of the study? The conclusion must have:

Remember the objective of the study; Main findings; Theoretical and practical implications; Originality of the study; Study limitations; Future lines of research

Reviewer 2 Report

Comments and Suggestions for Authors

I liked your work, however I think you missed a few important experiments/comments:

(1) what happens when an ISR is running and you are updating that specific code?

(2) what happens when you update non-ISR  code and and ISR is triggered?

(3) what happens when you update non-ISR code and that non-ISR code is running?

You have a comment like: - This process must be executed sequentially and without interruptions. If the sequence for some reason is interrupted the writing or erasing process is cancelled. << Please clarify if these interruptions also cover interrupts or not.

The overall remark from me is that you took an easy use-case where you only update non-ISR code, when that non-ISR code is not running. In addition for this particular case, I would also be interested what happens when you do the update while the Button1 is always kept pressed

Reviewer 3 Report

Comments and Suggestions for Authors

This paper presents a firmware update method during runtime. Compared to prior arts, the authors proposed a way of partial update of code blocks. Here are my comments:

1. Are the applications mentioned in the paper running in an RTOS or they are triggered by external interrupt? The task scheduling(if any) is not clear to the reader.

2. Would the update process block other applications? 

3. What if a power cycle happen during the flash writing? Would it be helpful to add read/verify after power up?

Round 2

Reviewer 1 Report

Comments and Suggestions for Authors

the authors made changes to the paper based on the reviewer's recommendations